# The Structural Cost of Anti-Aliasing in 3D Volumetric Segmentation

**Subhash Kashyap** 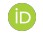                     124CS0013@NITRKL.AC.IN
*National Institute of Technology Rourkela*

## Abstract

Anti-aliasing techniques are commonly used to improve shift-equivariance in convolutional neural networks. While their benefits are well-established in 2D image classification, their role in 3D medical image segmentation remains unclear. In this work, we evaluate the effect of anti-aliasing by integrating BlurPool operations into a SegResNet architecture for brain tumor segmentation on the BraTS 2021 dataset (Menze et al., 2014; Bakas et al., 2018). The intervention reduces the Alias Violation Ratio (AVR) by approximately 50%, indicating effective suppression of high-frequency components. However, this reduction does not improve boundary-level segmentation performance and is accompanied by reduced shift consistency. These results suggest that high-frequency components, including aliased signals, may contribute to accurate boundary delineation in volumetric segmentation tasks.

**Keywords:** Spectral Aliasing, Volumetric Segmentation, Shift Equivariance, Brain Tumor, Deep Learning

## 1. Introduction

The Nyquist-Shannon sampling theorem states that signals should be band-limited prior to downsampling to avoid aliasing artifacts (Shannon, 1949). In convolutional neural networks, strided operations violate this condition, introducing high-frequency distortions into intermediate feature representations. In 2D image classification, anti-aliasing methods such as BlurPool have been shown to improve shift-equivariance and robustness (Zhang, 2019).

In contrast, the role of aliasing in dense prediction tasks such as 3D medical image segmentation is less well understood. These tasks require precise spatial localization, particularly at object boundaries. In clinical applications such as brain tumor segmentation, small boundary inaccuracies can significantly affect downstream decisions.

In this work, we investigate whether reducing spectral aliasing improves segmentation quality in a volumetric setting. We integrate anti-aliasing directly into a 3D SegResNet architecture (Myronenko, 2018) and evaluate its impact on spectral characteristics, boundary accuracy, and spatial robustness.

## 2. Methodology

### 2.1. 3D Spectral Tracking and Anti-Aliasing

To quantify spectral aliasing, we compute a 3D Fast Fourier Transform (FFT) of intermediate feature maps during inference. We define the Alias Violation Ratio (AVR) as the proportion of spectral energy lying outside the Nyquist limit:

$$\text{AVR} = \frac{\sum_{\Omega \notin Nyquist} P(u, v, w)}{\sum_{\text{All } \Omega} P(u, v, w)}$$

To enforce anti-aliasing, we replace strided downsampling operations with a 3D BlurPool kernel constructed via the outer product of a 1D binomial filter:

$$K_{3D} = \frac{1}{64} \left( \begin{bmatrix} 1 & 2 & 1 \end{bmatrix} \otimes \begin{bmatrix} 1 & 2 & 1 \end{bmatrix} \otimes \begin{bmatrix} 1 & 2 & 1 \end{bmatrix} \right)$$

This operation smooths feature maps prior to downsampling, reducing high-frequency components.

## 2.2. Boundary-Specific Evaluation

Standard Dice metrics are dominated by interior voxels and may not reflect boundary quality (Menze et al., 2014). We therefore evaluate segmentation performance using Boundary F1 (BF1), computed on thin morphological boundary regions derived from predicted and ground truth masks.

We also evaluate robustness using a Shift Consistency metric. Input volumes are translated spatially, and the Intersection over Union (IoU) between predictions from original and shifted inputs is measured. All experiments are conducted on $N = 251$ BraTS 2021 validation volumes.

## 3. Results and Discussion

Table 1: Evaluation metrics across 251 BraTS validation volumes.

| Architecture | Target Sub-Region | BF1 (%) | AVR (%) | 5px Shift IoU (%) |
|---|---|---|---|---|
| SegResNet Baseline | Enhancing Tumor | **72.60** | 6.61 | **98.0** |
| SegResNet BlurPool | Enhancing Tumor | 71.83 | **3.30** | 91.0 |

The BlurPool intervention reduces spectral aliasing substantially, lowering AVR from 6.61% to 3.30%. However, this reduction does not translate into improved segmentation performance. Boundary F1 decreases slightly (72.60% to 71.83%), with a small effect size (Cohen's $d = -0.087$), indicating limited practical impact. We note that the observed differences are not statistically significant, and should be interpreted as indicative trends rather than definitive effects.

Shift Consistency decreases under anti-aliasing, with IoU dropping from 98% to 91% at a 5-pixel translation. This indicates reduced robustness to spatial transformations. This is counterintuitive given that anti-aliasing is designed to improve shift equivariance; one possible explanation is that the blur operation degrades the spatial precision of feature representations, making predictions more sensitive to input perturbations rather than less.

Qualitative results (Figure 1) show that anti-aliasing produces smoother segmentation outputs, but with increased false positives around complex tumor boundaries. One possible explanation is that high-frequency components provide useful cues for resolving fine structural details, and suppressing them reduces boundary localization accuracy.

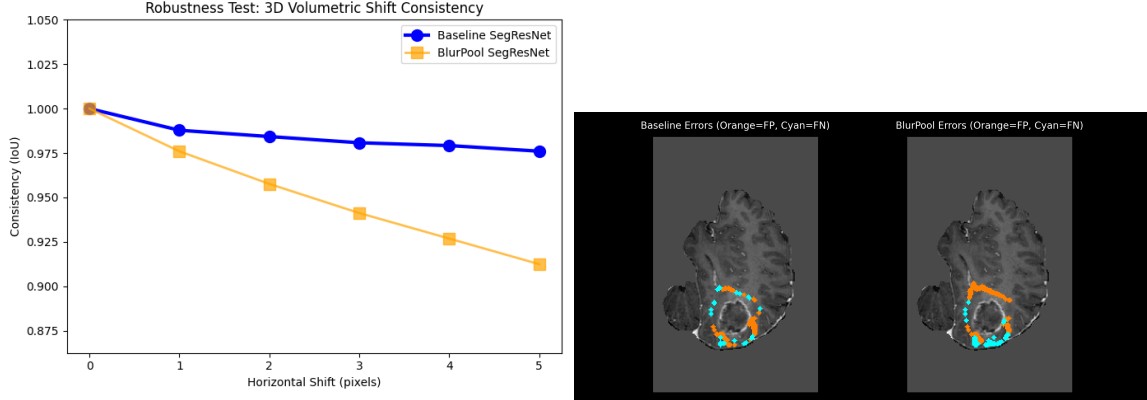

Figure 1: **Left:** Shift consistency curves for baseline and BlurPool models. **Right:** Error heatmaps showing false positives (orange) and false negatives (cyan).

## 4. Conclusion

We study the effect of anti-aliasing in 3D medical image segmentation and find that reducing spectral aliasing does not improve boundary-level performance. While BlurPool effectively suppresses high-frequency components, it may also remove information relevant for accurate boundary delineation. These findings highlight a potential trade-off between spectral regularization and spatial precision in volumetric segmentation architectures. Future work may explore adaptive or partial anti-aliasing strategies that preserve boundary-critical information while reducing harmful spectral artifacts.

**Code Availability**  Code and implementation details are publicly available at: https://github.com/Subkash2206/aliasing-tumor-boundaries.

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
