# OpenReview forum: "The Structural Cost of Anti-Aliasing in 3D Volumetric Segmentation"
_MIDL.io/2026/Short_Papers — MIDL 2026 - Short Papers Poster_

### Official Review · Reviewer_DHMw · 2026-05-04
**Revisiting anti-aliasing in 3D medical image segmentation**

**Rating:** 3
**Confidence:** 5

**Review:**

Overall, this paper provides a clear and well-executed empirical study on the role of anti-aliasing in 3D segmentation. The finding that reducing aliasing does not improve (and may even harm) boundary accuracy is interesting and somewhat counterintuitive.

Strengths:
- Clear experimental design and well-defined metrics (e.g., AVR, BF1)
- Insightful observation on the trade-off between spectral smoothing and spatial precision
- Relevant problem for medical image segmentation

Weaknesses:
- Contribution is mainly empirical with limited methodological novelty
- Experiments are restricted to a single architecture and dataset
- Performance differences are small and not statistically significant
- Limited analysis of underlying mechanisms (why anti-aliasing hurts performance)
- No comparison with alternative anti-aliasing or adaptive strategies

Overall, the paper offers useful observations but would benefit from broader validation and deeper analysis.

**Summary:**

This paper evaluates the impact of anti-aliasing (via BlurPool) in 3D medical image segmentation using a SegResNet architecture on the BraTS 2021 dataset. The authors introduce the Alias Violation Ratio (AVR) to quantify spectral aliasing and analyze its relationship with segmentation performance, boundary accuracy, and shift consistency. While anti-aliasing significantly reduces AVR, it does not improve boundary-level performance and slightly degrades both Boundary F1 and shift consistency. The study suggests that high-frequency components, including aliased signals, may be beneficial for precise boundary delineation in volumetric segmentation.

**Strengths:**

The paper addresses an underexplored question in 3D medical image segmentation and provides a clean empirical evaluation. The introduction of AVR as a spectral metric is useful, and the use of boundary-specific evaluation (BF1) is appropriate for the task. The results are consistent and highlight an interesting trade-off between reducing aliasing and preserving boundary detail. The study is well-structured and easy to follow.

**Weaknesses:**

The main limitation is the narrow experimental scope, as the study is conducted on a single model (SegResNet) and dataset (BraTS), limiting generalizability. The observed performance differences are small and not statistically significant, reducing the strength of the conclusions. The work lacks deeper analysis of why anti-aliasing degrades performance and does not explore alternative or adaptive anti-aliasing strategies. Overall, the contribution is primarily observational and somewhat incremental.

**Justification Of Rating:**

The paper presents a clear and interesting empirical finding, but the contribution is limited in scope and impact. The lack of broader validation and deeper analysis makes it difficult to assess general significance. With additional experiments and stronger insights, the work could be more compelling.

---

### Decision · Program_Chairs · 2026-05-08

Accept (Poster)